# Human Decidual CD1a^+^ Dendritic Cells Undergo Functional Maturation Program Mediated by Gp96

**DOI:** 10.3390/ijms24032278

**Published:** 2023-01-23

**Authors:** Tamara Gulic, Gordana Laskarin, Lana Glavan, Tanja Grubić Kezele, Herman Haller, Daniel Rukavina

**Affiliations:** 1Department of Physiology and Immunology, Faculty of Medicine, University of Rijeka, B. Branchetta 20, 51000 Rijeka, Croatia; 2Department of Obstetrics and Gynecology, Clinical Hospital Rijeka, University of Rijeka, Kresimirova 42a, 51000 Rijeka, Croatia; 3Department of Microbiology, Clinical Hospital Rijeka, University of Rijeka, Kresimirova 42a, 51000 Rijeka, Croatia; 4Department of Biomedical Sciences in Rijeka, Croatian Academy of Sciences and Arts, R. Matejčić 2, 51000 Rijeka, Croatia

**Keywords:** decidual dendritic cells, chemokine, cytokine, gp96, pregnancy

## Abstract

Heat shock proteins (hsps), in certain circumstances, could shape unique features of decidual dendritic cells (DCs) that play a key role in inducing immunity as well as maintaining tolerance. The aim of the study was to assess the binding of gp96 to Toll-like receptor (TLR) 4 and CD91 receptors on decidual CD1a^+^ DCs present at the maternal-fetal interface in vitro as well as the influence of CD1a^+^ DCs maturation status. Immunohistology and immunofluorescence of paraffin-embedded first-trimester decidua tissue sections of normal and pathological (missed abortion MA and blighted ovum BO) pregnancies were performed together with flow cytometry detection of antigens in CD1a^+^ DCs after gp96 stimulation of decidual mononuclear cells. Gp96 efficiently bound CD91 and TLR4 receptors on decidual CD1a^+^ DCs in a dose-dependent manner and increased the expression of CD83 and HLA-DR. The highest concentration of gp96 (1000 ng/mL) increased the percentage of Interferon-γ (INF-γ) and IL-15 expressing gp96^+^ cells. Gp96 binds CD91 and TLR4 on decidual CD1a^+^ DCs, which causes their maturation and significantly increases INF-γ and IL-15 in the context of Th1 cytokine/chemokine domination, which could support immune response harmful for ongoing pregnancy.

## 1. Introduction

At the maternal-fetal interface of early pregnancy, extensive tissue remodeling is necessary for providing the required maternal support and protection of the developing conceptus. Extensive tissue remodeling of decidual tissue is critical for successful implantation and is always associated with cellular stress response, damage, and necrosis of different cells subpopulation [1] with subsequent hsps increased in the intercellular space by passive release [2] or active secretion from living cells in response to damage [3]. The physiological functions of gp96, a member of the Hsp90 family [1], are common to other hsps, including proper cellular spatial proteins’ conformation and association, proteins degradation, signal transmissions and cellular survival, and proliferation by affecting apoptosis [4,5].

In normal early pregnancy, different hsps (Hsp90, Hsp27, Hsp70, and Hsp60) were found to be expressed by decidual stromal cells and endometrial glandular cells [6]. In addition, T. Gulic et al. (2013) [7] showed that trophoblasts, glandular epithelial cells, and cells dispersed in the decidual stroma express gp96 at the implantation site of early normal and pathological pregnancies (BO and MA). The gp96 expression levels were lower in retained pathological pregnancies (BO and MA) compared to normal first-trimester pregnancies [7], while on the systemic level, gp96 expression did not change significantly between normal pregnancy and women presenting threatened miscarriage of the same gestation age [8]. However, locally expressed gp96 could contribute to unwanted pregnancy outcomes in recurrent spontaneous abortion depending on the immunological status, as in tumors and autoimmune disease [7,8], due to the current understanding that outside of cells, gp96 could act as an alarmin by inducing a strong immunogenic activity [3,9]. One study showed that chronic infection of the upper parts of the genital tract with Chlamydia trachomatis leads to prolonged exposure of infected tissue to bacterial Hsp60 with the possibility of the development of autoimmunity to one’s own Hsp60 [10]. Therefore, women undergoing in vitro fertilization with a positive finding on the anti-chlamydial antibodies in the cervix have a much lower level of conception than women who do not have such antibodies. The mother’s serum containing the anti-Hsp60 markedly impaired mouse embryo development in vitro [10]. The formation of Hp60-antibody complexes and gp96-antibody complexes in the placenta may contribute to the induction of preterm birth [11]. In addition, the intraamniotic presence of anti-Hsp70 antibodies has been positively correlated with proinflammatory TNF-alpha and IFN-alpha cytokine concentrations [12]. The Hsp70 caused maturation of decidual CD1a^+^ DCs and IL-15 production in the context of Th1 cytokine/chemokine domination, which could support an immune response harmful to ongoing pregnancy [13,14]. Decreased gene and protein levels of interleukin-15 (IL-15) were found at the implantation site of pathological pregnancies [14].

The orchestration of adaptive immunity in vertebrates depends upon DCs, a class of bone-marrow-derived cells found in the blood, epithelia, and lymphoid tissues. DCs are equipped with molecular sensors and antigen-processing machinery to recognize pathogens, integrate chemical information, and guide the specificity, magnitude, and polarity of immune responses [1,15,16]. A functional–anatomical classification derived from murine studies recognizes that DCs function is intimately linked to location [17,18]. In tonsils, lymph nodes and spleen DCs are equipped with a wide range of lectins, toll-like receptors (TLRs) and other pattern recognition receptors for the general purpose of antigen uptake, transport, and presentation compared to decidua of the first trimester of normal pregnancy where the majority of DCs present mature tolerogenic phenotype CD83^+^ DCs and only around 1–2% are immature myeloid CD1a^+^ DCs [19,20,21]. Immature CD1a^+^ DCs are a tiny population in decidual tissue, mostly expressed CCR5^+^ (CD195^+^), to be kept by locally produced pro-inflammatory chemokines CC-ligand 3 CCL3 [16]. This CD1a^+^ cell population also expressed pattern recognition receptors, such as Toll-like receptor (TLR) 4 [7], CD91 [7], CD205 [17], the mannose receptor (MR) or CD206 [6], and CD209 [7,8]. This phenotype provides decidual immature DCs the possibility to recognize environmental antigens as potent modulators of their functions and act as sentinels at the implantation site [21,22]. DCs could internalize gp96 alone or in the complex with cell peptides by receptor-mediated endocytosis [13] and loaded to the MHC class I and II molecules in order to prime T and/or B cells shaping immunological response [4,13]. In a few studies, hsps (Hsp70, Hsp60, and Hsp90) have been considered the potential reason for causing abortion [10,14,23]. However, the function of decidual DCs in the human decidua is still far from being understood [9,14]. 

Increased local levels of extracellular hsps may be one of the possible factors involved in altering placental function in vivo [24,25,26,27]. In this context, decidual DCs seems to be an interesting target for investigation due to their inherent plasticity to bias immune response upon stimulation with particular hsps. The effect of gp96 on the immunobiology of decidual DCs during early normal and pathological pregnancies is still unknown.

The aim of this study was to assess the binding of gp96 for TLR4 and CD91 receptors on decidual CD1a^+^ DCs in vitro and to analyze the influence on CD1a^+^ DCs activation status and maturation process. The results indicate that gp96 specifically binds CD91 and TLR4 and causes a proinflammatory maturation program in normal early pregnancy decidual CD1a^+^ DCs mediated by increased production of INF-γ and IL-15 in the context of Th1 cytokine/chemokine domination. 

## 2. Results

### 2.1. Cytokeratin Expression Pattern in Decidua of Normal and Pathological Pregnancies

Immunohistology-labeled sections of cytokeratin-positive (cytokeratin^+^) cells were detected in extravillous trophoblast cells, trophoblast villi, and glandular cells, of NP (Figure 1A), BO (Figure 1B) and MA (Figure 1C). Quantification of the immunohistological labels showed that the number of cytokeratin^+^ cells per mm^2^ was significantly lower in MA compared to NP and BO, while the staining intensity (H score) for cytokeratin in decidua does not differ between normal and pathological pregnancies (Figure 1J). However, the frequency of cytokeratin^+^ trophoblast cells was significantly lower in trophoblast cells of pathological pregnancies (MA and BO) compared to NP, but the intensity of cytokeratin labeling (H score) did not change among the investigated groups (Figure 1).

### 2.2. Assessment of Cell Proliferation Marker Ki-67 at the Maternal–Foetal Interface

Immunohistology-labeled sections of Ki-67-positive (Ki-67^+^) cells were randomly distributed in the decidual stroma of NP (Figure 2A), BO (Figure 2B), and MA (Figure 2C). In normal and pathological pregnancies, Ki-67 is detected in the nucleus of extravillous trophoblast cells, in some cells lining the glands, and in cells dispersed in the stroma (Figure 2A, black arrows). Specifically labeled Ki-67^+^ cells (brown) were more abundant in the first trimester of NP pregnancy decidua (Figure 2A, upper row) than in B.O (Figure 2B, upper row) and M.A (Figure 2C, upper row). The co-expression of Ki-67 (red fluorescence) and cytokeratin (green fluorescence) was not detected in the cytoplasm, while the co-expression of Ki-67 and DAPI fluoresces purple in the nuclei in all investigated groups (Figure 2A–C lower row). Some decidual cells are single labeled with anti-Ki-67 in all samples (Figure 2A–C lower row, white arrow). Moreover, glandular cells and infiltrating trophoblast cells in all decidual sections were Ki-67^−^ and cytokeratin^+^ cells (green fluorescence) (Figure 2A–C, lower row, arrowheads). In addition, only some rare extravillous trophoblast cells and cells lining the glands were found Ki-67^+^ (red fluorescence)/cytokeratin^+^ cells (green fluorescence) (Figure 2A–C). 

### 2.3. Gp96 Specifically Bind CD91 and TLR4 Receptors Expressed on Decidual CD1a^+^ Immature Dendritic Cells 

The frequency of the CD1a^+^ DCs in freshly isolated DMC suspension was analyzed by flow cytometry within the R1 gate set for DCs positive events (back gating using CellQuest Pro Software, Beckton Dickinson) (Figure 3A). This gate comprises larger cells of more complex granularity than the characteristic lymphocyte gate.

In the region, R2 gate set up for CD1a^+^ events 3.7% of cells were found among the cells belonging to gate R1. Numbers in the histograms illustrate the percentage of CD91 expressing CD1a^+^ cells (Figure 3B) and TLR4 (Figure 3C) expressing CD1a^+^ cells belonging to gates R1 and R2 cultured in the medium only in the presence of gp96. Percentages of CD91 and TLR4 expressing CD1a^+^ DCs decreased significantly with the increase in gp96 concentrations (Figure 3B, ** *p* = 0.001 and 3C, * *p* = 0.03). 

### 2.4. Gp96 Include Maturation of Decidual CD1a^+^ DCs 

Among investigated receptors, only the frequency of maturation CD83 marker increased in CD1a^+^ cells at the concentration of 1000 ng/mL of gp96 (* *p* = 0.002, Figure 4A) as well as MFI for HLA-DR at the concentration of 100 ng/mL (* *p* = 0.01, Figure 5B). Gp96 did not significantly affect the percentage of HLA-DR, CD80, or CD86 markers on CD1a^+^ cells (Figure 5A). Histograms illustrate the analyses (Figure 4C and Figure 5C).

### 2.5. Gp96 Enhanced Expression of Pro-Inflammatory Cytokines on Decidual CD1a^+^ DCs 

Gp96 did not change the frequency of IL-4 expressing CD1a^+^ cells while efficiently enhancing the frequency of IL-15 (1000 ng/mL) and IFN-γ (100 and 1000 ng/mL) in CD1a^+^ cells (* *p* = 0.01, Figure 6A). Gp96 did not affect the expression of CCL3, CCL17, and CCL22 in CD1a^+^ cells compared to unstimulated cells (Figure 6B). 

### 2.6. IL-15 Upregulate the Expression of gp96 in DMCs from Early Normal Pregnancy Decidua

The expression of gp96 mRNA from freshly isolated DMCs of normal pregnancy was upregulated after 18 h stimulation with IL-15 by approximately four times compared to non-stimulated cells (Figure 7). 

## 3. Discussion

In early pregnancy, trophoblast cells proliferate, migrate, and invade the pregnant uterus in order to create appropriate local conditions for nourishing the developing embryo [15,17] in a very similar fashion as tumors [25]. During extensive tissue remodeling at the maternal-fetal interface of normal and pathological pregnancies, increased levels of intra and extracellular hsps were observed [7,14,27,28], where they can act as strong immunogens recognized by professional antigen-presenting cells [3,14]. Appropriate interaction between maternal immune cells and invading trophoblast cells during the first trimester is critical for the maintenance of pregnancy. Uterine natural killer (uNK) cells, immature dendritic cells (iDCs), T cells, and macrophages contribute to modulating the uterine environment for sustaining a successful pregnancy [16,20,21]. 

Pathological pregnancies of the first trimester, BO and MA, are characterized by trophoblast retention in the uterus before the outward signs of pregnancy termination appear. Aberrant behavior of DCs could affect trophoblast function and placental development, potentially leading to adverse pregnancy outcomes. Gulic et al. [7] showed the number of gp96^+^ cells decreased in BO and MA at the maternal-fetal interface, indicating a weaker functional activated status of immune cell response. For the first time, we evaluated the expression of cytokeratin as a marker for trophoblasts in normal and pathological early first-trimester pregnancies. Moreover, glandular cells and infiltrating trophoblast cells in all decidual sections were determined as cytokeratin^+^ cells, indicating the presence of the maternal-fetal interface. Interestingly, the absolute numbers of cytokratin^+^ cells were significantly lower in the decidual part of MA in comparison to NP and BO first-trimester pregnancy decidua. In concordance, a significantly lower number of cytokratin^+^ cells in villous trophoblast in BO and MA were detected in comparison to NP. It is likely that the influence of physiologically present growth factors, hormonal stimulation, and/or immunological status regulate tumor-like intensive trophoblast proprieties at the beginning of pregnancy could significantly support the expression of cytokratin^+^ cells in the decidua of normal and pathological pregnancies ex vivo [7]. Malihe Hasanzadeh et al. (2012) [26] showed that the majority of cells in cytotrophoblastic columns and shells have nuclear reactivity with proliferation marker Ki-67 and that Ki67 oncogene in trophoblastic cells in patients with gestational trophoblastic neoplasia is found far more frequently. In addition, Ki-67 was investigated at the implantation site in normal and pathological pregnancies, and its expression was found lower in pathological pregnancies, confirming our speculation.

We hypothesized that extracellular gp96, at the maternal-fetal interface of early pregnancy, could mediate extensive inflammatory immune reaction via DCs activation, as it was earlier shown in different tumors on local and systemic levels [7,29,30]. Dendritic cells (DCs) are the most potent antigen-presenting cells (APC) at the maternal-fetal interface, which may assist in various aspects of decidual homeostasis, placental development, and tolerance to the semi-allogeneic trophoblast [14,15]. Herein, we showed that gp96 efficiently binds CD91 and TLR4 receptors on decidual CD1a^+^ DCs in a dose-dependent manner in vitro. This observation is in line with the increasing evidence showing the involvement of gp96/CD91 or gp96/TLR4 interactions in the pathogenesis of different inflammatory diseases, such as infection [31,32], tumor [30], or allergic disorders [33]. Similarly, TLR2/4 and CD91 on bone marrow-derived mouse myeloid DCs interact via gp96 to up-regulate co-stimulatory molecules (CD80 and CD86) and MHC class II, thus enabling them for effective antigen presentation [19] and clonal expansion of cytotoxic T (CTL) and NK cells in vitro and in vivo [2,5]. In line with the literature, we found increased expression levels of CD83 and HLA-DR after stimulation with gp96 on decidual CD1a^+^ DCs in vitro. Fine-tuning of CD1a^+^ DCs maturation for effective antigen presentation resulting in reduced capacity to activate CD8^+^ T in the animal model. Thereby, this maturation status of CD1a^+^ DCs could possibly affect excessive immune reactions leading to rejection of trophoblast [34,35,36,37]. In addition, the interaction of gp96 with CD91 and TLR4 on DCs mediated activation of the NF-κB signaling pathway, increasing the expression of maturation markers and enhanced secretion of pro-inflammatory cytokines in vitro and in vivo [14,27]. Moreover, gp96 in our experiment did not alter the frequency of CCR5 and CCR7 receptors on decidual CD1a^+^ DCs, allowing their action locally. Gp96, in a dose-dependent manner, also increased remarkably the percentages of IL-15 and IFN-γ expressing CD1a^+^ DCs bias pro-inflammatory microenvironment domination. IL-15 is an important player in the decidualization process, including angiogenesis [38,39], accumulation of particularly shaped decidual uNK [40,41], and regulation of trophoblast invasion [7,39]. On the other hand, gp96 mRNA is upregulated in decidual mononuclear cells after IL-15 stimulation in vitro, but in our experimental model, gp96 and IL-15 expression significantly decreased in the decidual stroma of women with BO and MA [7]. Many of these IFNs are employed in normal pregnancy and development, as well as in defense against pathogens [40,41,42]. Aberrant expression of IFN-γ can alternate or induce aberrant activation of decidual NK cells during pregnancy that can lead to pregnancy complications. Our results indicate that an increased concentration of gp96 at the maternal-fetal interface may be one of the mechanisms that could be a trigger for increased production of INF-γ.

In fact, recently, we showed a decrease in authentic decidual NK cell number, and reduced activation of GNLY-mediated killing might be implicated in the slower rejection of trophoblast cells in BO and MA. This could be responsible for low cytotoxicity against trophoblast cells in MA. It may be linked to disturbed implantation and vascularization with consecutive placental and fetal retention in these pathological pregnancies [42]. 

In conclusion, each factor that increased the extracellular surplus of gp96 at the maternal-fetal interface, as the natural ligand for CD91 and TLR4 receptors, could support harmful immune response and compromise ongoing pregnancy by affecting the delicate maturation program in decidual CD1a^+^ DCs. Hopefully, further research on decidual DCs and their interaction between decidual DCs with T and NK cells at the maternal-fetal interface will shed light on how this delicate balance between tolerance gp96-mediated mechanisms and immunity to threatening agents expand our understanding of at implantation site is brought about and maintained in normal and pathological early pregnancy decidua. 

## 4. Materials and Methods

### 4.1. Human Tissue Samples

Decidual tissue samples were received from women with normal first-trimester pregnancies (NP, *n* = 30) and pathological pregnancies, which included BO (*n* = 25) and MA (*n* = 25). Normal pregnancy decidual tissues were obtained from women who underwent vaginal elective termination of a healthy pregnancy and had at least one live-born child. All women underwent a vaginal termination of 6-to 9-week-old pregnancies at the Department of Obstetrics and Gynecology, Clinical Hospital, University of Rijeka, Croatia. The women were 22–33 years old and did not have acute infections, chronic, degenerative, hereditary diseases, or previously known chromosomal abnormalities. All the women included in the study signed informed consent forms. The study was approved by the Ethics Committee of the Medical Faculty, University of Rijeka, Croatia, and all the women signed informed consent before tissue collection. The karyotypes of the abortive specimens were not analyzed. None of the women with BO and MA previously had a miscarriage, as observed in the medical documentation. The study was approved by the Ethics Committee of the Medical Faculty, University of Rijeka, Croatia, and informed consent was obtained from all women before receiving tissue samples.

### 4.2. Immunohistology of Paraffin Embedded Tissue Sections 

Tissue sections (6 μm) of paraffin-embedded early pregnancy decidual samples from normal and pathological pregnancies (BO and MA) were single-labeled for cytokeratin and Ki-67 detection, following the principle described earlier by Gulic and colleagues (2016) [42]. Briefly, the sections were deparaffinized in Tissue Clear (Paraffin Cleaning Agent, Sakura Fintek Europe, Zoeterwoude, Netherland), hydrated in decreasing concentrations of ethanol (from 100% to 75%), and washed in phosphate-buffered saline (PBS; 0.05 M containing 0.3 M NaCl; pH 7.4) (Kemika, Zagreb, Croatia). All the incubations were performed at room temperature (RT). Antigens were retrieved by boiling in 10 mM sodium citrate (pH 6.0) in a microwave oven and sections washed in PBS. Endogenous peroxidase activity was inhibited with 3% H_2_O_2_ (Kemika, Zagreb, Croatia) for 10 min. Non-specific antibody binding was prevented by incubation with 2% bovine serum albumin (BSA; Sigma Aldrich Chemie, Steincheim, Germany) for 45 min. The sections were then incubated overnight at +4 °C with primary anti-Cytokeratin or anti-Ki-67 antibodies of interest or antibodies of irrelevant specificity, which served as isotype-matched controls (all specified in Table 1). After washing the sections in PBS, secondary biotinylated antibodies (specified in Table 1) were added and incubated for 1 h with streptavidin peroxidase (to cover a slide) (Zymed Laboratories, Invitrogen Corporation, Carlsbad, CA, USA) was subsequently added to the slides for 20 min. The binding of antibodies to the specific binding sites was visualized using 3, 3-diaminobenzidine (DAB; Dako, Hamburg, Germany), while the nuclei were stained with hematoxylin (Gill No. 3 solution, Sigma Aldrich Chemie, Steincheim, Germany). Slides were analyzed with an Olympus BX51 light microscope (Tokyo, Japan). Microphotographs were acquired using a figureOlympus DP71 camera and Cell^an imaging software, version 3.0 (both from Olympus, Tokyo, Japan). Magnification was achieved by the Olympus UPlan objective lens, FI 10×/0.3 (100×); FL 40 ×/0,75∝/0.17/FN 26.5 (400×); Apo100×/1.35 Oil Iris (1000×). 

The immunohistological labeling was quantified using the Alphelys Spot Browser 2 integrated system (Alphelys, Plaisir, France), as described in detail in our previous publication [7,42]. The algorithms were designed to determine the count of unstained and stained objects (cells), and the results are expressed as the arithmetic mean of the number of positive cells/mm^2^ of the tissue analyzed in 12 fields (magnification 400×). The software detected objects (cells) of interest on the basis of pixel color properties (wavelength, intensity, and saturation), and the results are expressed as an ”H score” and represent the intensity of labeling.

### 4.3. Immunofluorescence of Paraffin Embedded Tissue Sections

Immunofluorescence was performed on paraffin-embedded early pregnancy decidual tissue sections (6 µm) from normal and pathological pregnancies (BO and MA), according to the protocol described earlier [7]. Sections were deparaffinized, hydrated, and washed, antigens were retrieved, and non-specific antibody binding prevented at RT, as it was described above for the immunohistology method. Tissue sections were subsequently incubated over night at +4 °C with anti-cytokeratin/anti-Ki-67. The control labeling was performed with isotype-matched antibodies. All the antibodies are specified in Table 1. After washing in PBS, secondary antibodies conjugated with fluorescent dies (all specified in Table 1) were added to cells for 40 min at RT. The nuclei were stained with 4, 6-diamidino-2-phenylindole (DAPI; blue fluorescent dye) (Sigma Aldrich Chemie, Steincheim, Germany). After washing with PBS, slides were mounted with Mowiol medium (Sigma Aldrich Chemie). Fluorescent images were acquired with an Olympus BX51 fluorescent microscope using an Olympus DP71 camera and Cell^A imaging software, version 3.0 (both from Olympus, Tokyo, Japan). Magnification was achieved by the Olympus UPlan objective lens, FI 10×/0.3 (100×); FL 40 ×/0.75∝/0.17/FN 26.5 (400×); Apo100×/1.35 Oil Iris (1000×). In double-labeled tissue sections, the acquired images were overlaid using Adobe Photoshop, version 7.0.1 CE (Adobe Systems Inc., San Jose, CA, USA).

### 4.4. Isolation of Decidual Cells by Enzymatic Digestion

Decidual mononuclear cells (DMCs) were isolated from tissue samples of the first trimester of normal human pregnancy decidua, in vitro, according to the method established in our laboratory [18,20,22]. Decidual tissue was washed, cut into pieces, and exposed to collagenase IV enzyme (equal volume of tissue and 0.1% collagenase type IV; Sigma, Munchen, Germany) at 37 °C for 45 min with gentle stirring using a magnetic stirrer (Ikamag, IKA, Staufen, Germany). Tissue debris was eliminated from the suspension by passing twice through a 100-μm nylon mesh (TTP, Trasadingen, Switzerland), and the cells were centrifuged at 350× *g* for 10 min. The cell pellet was resuspended in RPMI 1640, overlaid onto the Lymphoprep solution (Axis-Shield PoC AS, Oslo, Norway), and centrifuged at 600× *g* for 20 min without brake centrifuge option. The cells were carefully collected by pipet. After washing in RPMI 1640, the cells were resuspended in RPMI 1640 containing 10% fetal calf serum (FSC; Gibco, Gaithersburg, MD, USA), 0.3 mg/mL L-glutamine and 100 U/mL penicillin/streptomycin and adjusted to a concentration of 10^6^ cells/mL. Cell viability was always >95%, as it was analyzed using trypan blue (Serv, Heidelberg, Germany) and a light microscope (Carl Zeiss, Jena, Germany).

### 4.5. Binding Assay 

The binding of gp96 for CD91 AND tlr4 was assessed in freshly isolated DMC from normal first-trimester early pregnancy. DMC (2.5 × 10^5^ cells/sample) were untreated or treated with gp96 (Sigma Aldrich Chemie, Steincheim, Germany) at a concentration of 0.125 µg/mL, 0.25 µg/mL, 0.5 µg/mL, and 1 µg/mL for 30 min on ice. Then fluorescein isothiocyanate (FITC), either conjugated anti-CD91 mAb (IgG_2b_) or TLR4 (IgG_1_) mAb diluted 1:100 in fluorescence-activated cells sorter (FACS) buffer [NaCl (8.12 g/L), KH_2_PO_4_ (0.26 g/L), Na_2_HPO_4_ (2.35 g/L), KCl (0.28 g/L), all from Kemi-ka, Zagreb, Croatia; Na_2_EDTA (0.36 g/L), Fluka, Buchs, Switzerland; NaN3 (0.1 g/L)] was added into the samples for 30 min. FITC conjugated mouse IgG1 or FITC conjugated mouse IgG2b, respectively, were used as isotype controls. After washing in FACS buffer, CD1a was labeled using PE-conjugated anti-CD1a mAb added, respectively (30 min. at +4 °C). 

### 4.6. Antigen Detection in DMCs Using Flow Cytometry

Freshly isolated DMC that had been cultured for 18 h at 37 °C in a humidified 5% CO_2_ incubator in tissue culture Petri dishes (TTP, Trasadingen, Switzerland) (10 × 200 mm) just in 10% FSC RPMI 1640 medium or in the presence of gp96 at a concentration of 1 ng/mL, 100 ng/mL and 1000 ng/mL. They were stained for activation markers, chemokine receptors, or intracellular cytokine and chemokine detection. Cell samples, in which intracellular cytokine and chemokine detection was performed, had been initially incubated with phorbol-myristate-acetate (6M), ionomycin (1M), and monensin (3M) for 5 h at 37^◦^C in a humidified atmosphere containing 5% CO_2_ prior to cytokine and chemokine staining. The labeling of cytokines and chemokine in decidual DCs was performed according to the method described earlier [20,22]. Briefly, the cells were washed in FACS buffer and fixed with 4% paraformaldehyde (Kemika, Zagreb, Croatia) for 10 min at RT. After 2 washes in cold FACS buffer, unspecific binding was blocked by pre-incubation with fetal calf serum (FCS) for 20 min at room temperature. The cells were double-stained with FITC-conjugated anti-CD1a mAb and PE conjugated antibody specific for CD83, CD80, CD86, HLA-DR, IL-4, IFN-γ, IL-15, CCL3, CCL17 and CCL22. Directly conjugated (FITC or PE) mouse isotype-matched immunoglobulins were used as isotype controls. All mentioned antibodies are described in Table 1. After labeling (30 min at +4 °C), samples were washed twice in cold FACS buffer, fixed in 200 µL 2% paraformaldehyde, and assessed by flow cytometry (Becton Dickinson FACS Calibur) using CellQuestPro software (BD Biosciences, San Jose, California, CA, USA).

### 4.7. RNA Preparation and Real-Time Quantitative Polymerase Chain Reaction

Total RNA was isolated from freshly isolated DMCs of normal pregnancies using the TRIzol method (Invitrogen, Carlsbad, CA, USA) that had been cultured for 18 h at 37 °C in a humidified 5% CO_2_ incubator in tissue culture Petri dishes (TTP, Trasadingen, Switzerland) (10 × 200 mm) just in 10% FSC RPMI 1640 medium or in the presence of IL-15 at a concentration of 2 ng/mL and 4 ng/mL. The quality of RNA samples was examined by agarose gel electrophoresis (sharp 18S and 28S ribosomal RNA bands) and UV spectrophotometry (high OD260/OD280 ratio). The samples were treated with the Turbo DNA-free kit (Ambion Inc, The RNA Company, Austin, TX, USA) to eliminate any traces of DNA contamination. Subsequently, RNA samples having a high quality (5 μg) were reverse-transcribed (High Capacity cDNA Archive Kit; Applied Biosystems, Foster City, CA, USA) according to the protocol supplied by the kit manufacturer. SDS (Applied Biosystems) and commercially available TaqMan assay reagents (Applied Biosystems) for Gp96 (Hs00427665 g1) and 4352930E (18S rRNA) (Applied Biosystems), according to the manufacturer’s instructions. Briefly, RT-qPCR analysis was performed in 25 μL of the reaction mixture using gene-specific primers, 5 μL of cDNA as a template, and the TaqMan Universal PCR Master Mix (Applied Biosystems). RT-qPCR was performed on the Applied Biosystems 7300 real-time PCR system. To compensate for possible inter-PCR variation, normalization of the target genes with an endogenous control gene (18S rRNA) was performed in each experiment. The results are expressed as 2^ΔΔCt^ values, calculated as the difference between the ΔΔCt values of DMCs of pathological pregnancies and the DMCs of normal pregnancies. RT-qPCR results are presented relative to those of the DMCs of the first trimester of normal pregnancy, which was assigned a value of 1.

### 4.8. Statistical Analysis

The data are expressed as median and (25th and 75th percentiles). The significance of the differences between multiple groups of interest was analyzed using the Kruskal-Wallis non-parametric test, and the difference was significant at *p* < 0.05. Dunn’s test was used as a post hoc test to establish among which group the difference existed with a level of significance adjusted to the number of mutual comparisons. Statistical analyses were performed with the data analysis software system Statistica 7.0 (StatSoft, Inc., Tulsa, OK, USA).

## Figures and Tables

**Figure 1 ijms-24-02278-f001:**
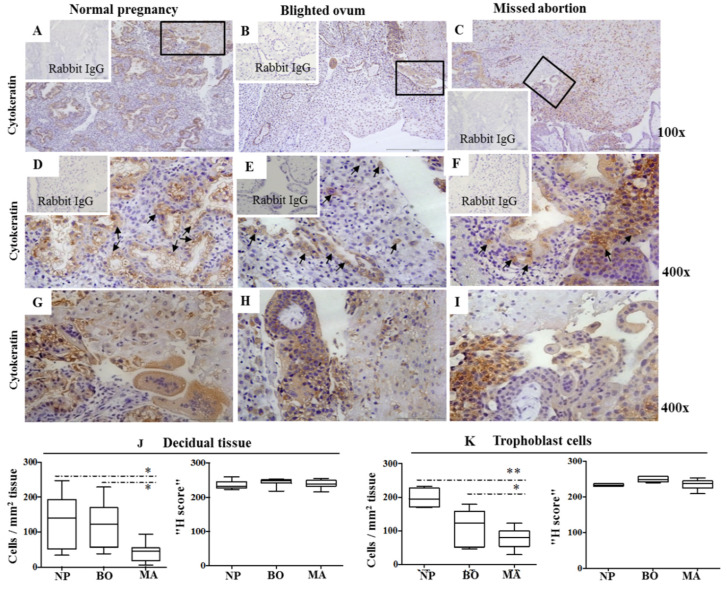
Cytokeratin expression in the decidua of normal pregnancy (**A**,**D**,**G**), blighted ovum (**B**,**E**,**H**), and missed abortion (**C**,**F**,**I**). Immunohistological labeling (**A**–**F**) using mouse IgG1 anti-Cytokeratin mAbs is visualized in brown color using 3,3-diaminobenzidine. Details from the lined quadrangles ((**A**–**C**), 100×) are shown at higher magnification ((**D**–**I**) 400×). Black arrows show the sites of trophoblast invasion (**D**–**F**). Graphs show the number of Cytokeratin-positive cells scattered per mm^2^ and the H score for Cytokeratin in the decidual stroma (**J**) and trophoblast cells (**K**) using immunohistology (NP—normal pregnancy; BO—blighted ovum; MA—missed abortion). Ten samples were analyzed per group. Levels of significance: * *p* = 0.016 and ** *p* = 0.0078.

**Figure 2 ijms-24-02278-f002:**
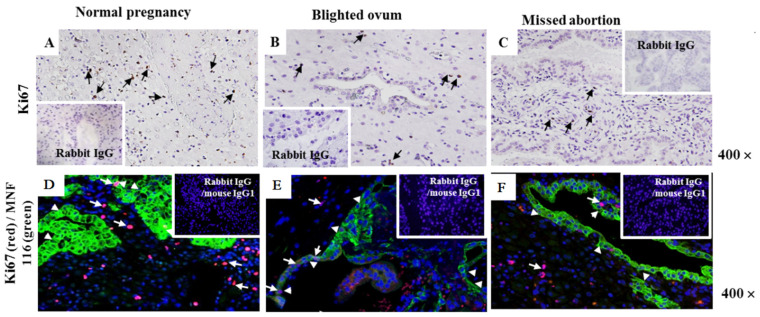
Ki-67 and Ki-67/cytokeratin co-expression in the decidua of normal pregnancy (**A**,**D**), blighted ovum (**B**,**E**), and missed abortion (**C**,**F**). Immunohistology (**A**–**C**) shows single labeling with mouse anti-Ki-67 mAb or mouse IgG1 (inserts) using indirect immunoperoxidase staining. Ki-67^+^ cells are stained brown (black arrows). Double immunofluorescent labeling of Ki-67 and Cytokeratin in NP (**D**), BO (**E**), and MA (**F**). Ki-67 fluoresces red, and Cytokeratin fluoresces green. The nuclei are stained blue by DAPI. Purple fluorescence visualizes the merge of red and blue color (white arrows), while green fluorescence (white arrowheads). The inserts represent isotype-matched control. Ten samples were in each group. Magnification: 400×.

**Figure 3 ijms-24-02278-f003:**
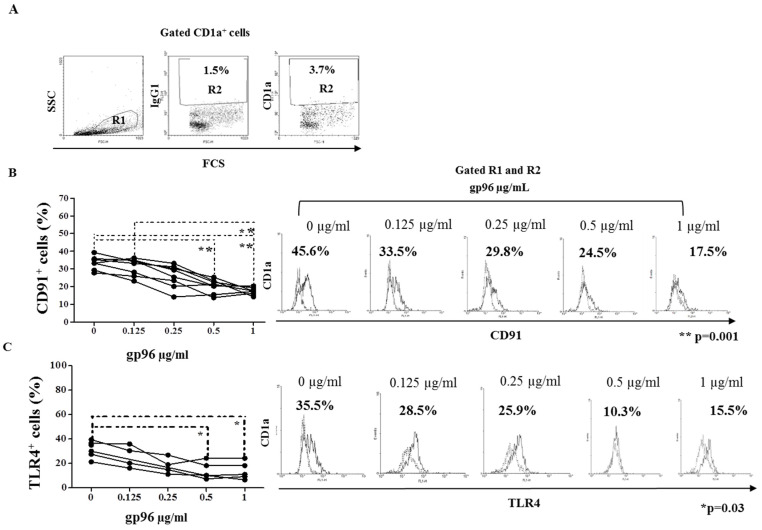
Specific binding of gp96 for CD91 and TLR4 on CD1a^+^ dendritic cells from the suspension of early pregnancy decidual mononuclear cells. Dot plots show the gating of decidual CD1a^+^ cells within the region (R) 1 (back gating for CD1a^+^ events) and R2 gate (CD1a^+^ events) with respect to isotype mouse IgG1 (**A**). The results of binding assays are shown as the percentages of CD91 (**B**) or TLR4 (**C**) expressing CD1a^+^ cells calculated as the difference compared to isotype-matched controls after the treatment with gp96 at the indicated concentrations. Histograms illustrate the labeling of CD1a^+^ cells with antibodies of interest (solid histogram curves) in relation to isotype-matched control (dashed histogram curves). Levels of statistical significance: ** *p* = 0.001 and * *p* = 0.03.

**Figure 4 ijms-24-02278-f004:**
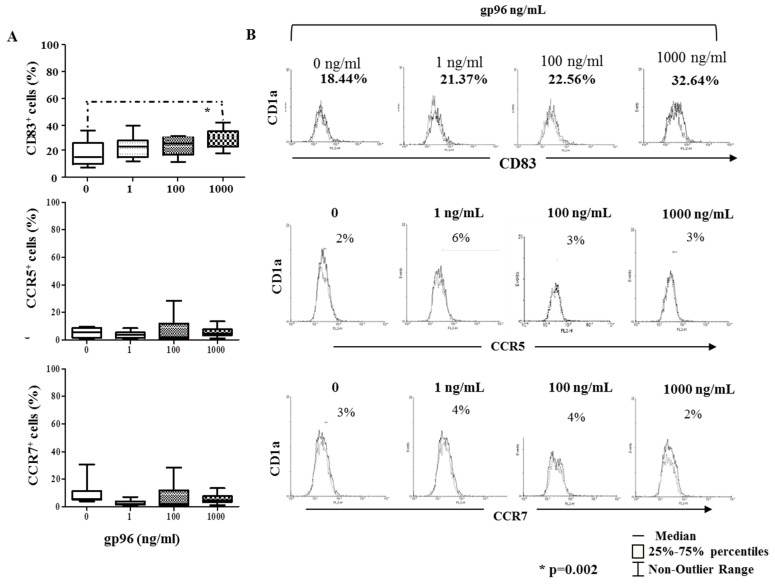
The percentages (**A**) for the cell surface molecules CD83, CCR5, and CCR7 on CD1a^+^ cells from the suspensions of early pregnancy decidual mononuclear cells after the 18 h-culture in the medium only or with gp96 at indicated concentrations. Histograms (**B**) illustrate the labeling of CD1a^+^ cells with antibodies of interest (solid histogram curves) with respect to isotype-matched control (dashed histogram curves). The results are expressed as the median of 7–8 independent experiments performed in each group. Level of statistical significance: * *p* = 0.002.

**Figure 5 ijms-24-02278-f005:**
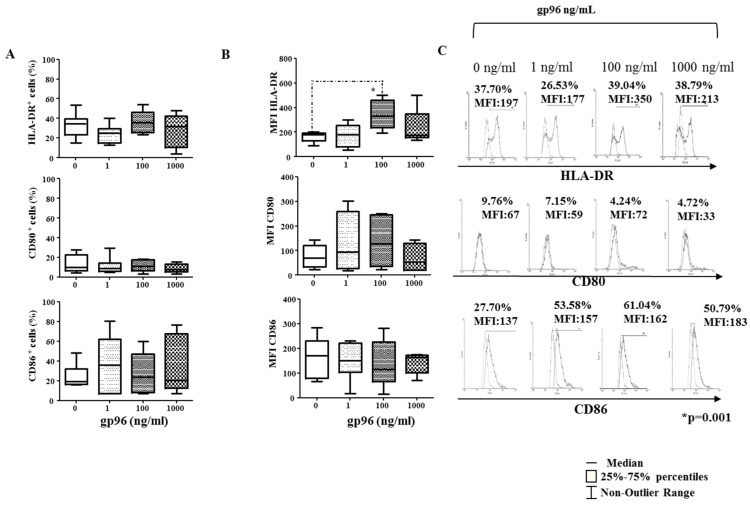
The percentages (**A**) and Mean Fluorescence intensity (MFI, (**B**)) for the cell surface molecules CD80, CD86, and HLA-DR on CD1a^+^ cells from the suspensions of early decidual mononuclear cells after the 18 h-culture in the medium only or with gp96 at indicated concentrations. Histograms (**C**) illustrate the labeling of CD1a^+^ cells with antibodies of interest (solid histogram curves) in relation to isotype-matched control (dashed histogram curves). The results are expressed as the median of 7–8 independent experiments performed in each group. Levels of statistical significance: * *p* = 0.001.

**Figure 6 ijms-24-02278-f006:**
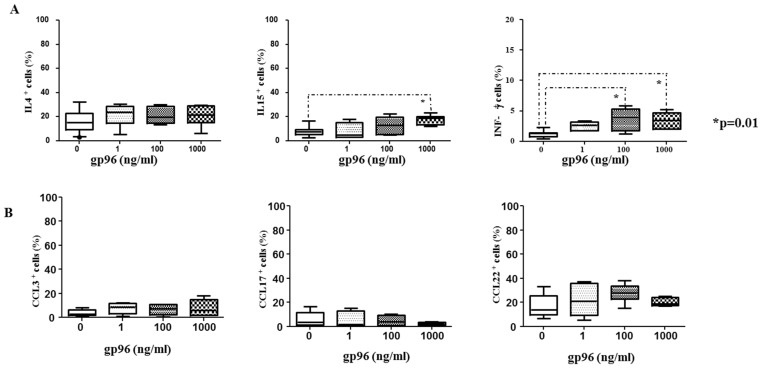
The percentages of intracellular cytokines interleukin (IL)-4, interferon-gamma (IFN-γ) or IL-15 and chemokines CCL3, CCL17, or CCL22 expressing CD1a^+^ cells from the suspensions of early decidual mononuclear cells after the 18 h—culture in the medium only or with gp96 at indicated concentrations are shown in the graphs (**A**,**B**). 7–8 independent experiments were performed in each group. Levels of statistical significance: * *p* = 0.01.

**Figure 7 ijms-24-02278-f007:**
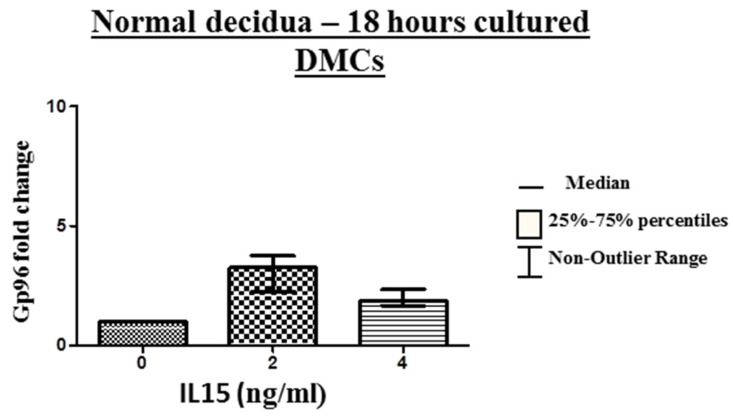
Gp96 mRNA expression in freshly isolated decidual mononuclear cells (DMCs) of normal pregnancy (NP), blighted ovum (BO), and missed abortion (MA). The results are expressed as fold changes of gp96 mRNA levels of BO and MA relative to vehicle control.

**Table 1 ijms-24-02278-t001:** Specification of antibodies used.

Primary Antibodies			
Specificity	Clone	Provider	Concentration
Cytokeratin	MNF 116 (mouse IgG1)	DAKO, Carpinteria, CA, USA	1:100
Cytokeratin	Polyclonal (rabbit IgG)	Abcam, Cambridge, UK	1:100
Ki-67	SP6 (rabbit IgG1)	Abcam, Cambridge, UK	1:100
CD1a (FITC, PE)	HI149 (mouse IgG1)	BD Biosciences, San Diego, CA, USA	20 μL/10^6^ cells
TLR4 (FITC)	76B357.1(mouse IgG2b)	Abcam, Cambridge, UK	2.5 μg/10^6^ cells
CD91 (PE)	A2MR-α2 (mouse IgG1)	BD Biosci. San Diego, CA, USA	20 μL/10^6^ cells
CD83 (PE, PE-Cy5)	HB15e (mouse IgG1)	BD Biosciences, San Diego, CA, USA	20 μL/10^6^ cells
CCR5 (PE)	2D7 (mouse IgG2a)	BD Biosciences, San Diego, CA, USA	20 μL/10^6^ cells
CCR7 (PE)	3D12 (rat IgG2a)	BD Biosciences, San Diego, CA, USA	20 μL/10^6^ cells
CD80 (PE)	L307.4 (mouse IgG1)	BD Biosciences, San Diego, CA, USA	20 μL/10^6^ cells
C86 (PE)	IT2.2 (mouse IgG1)	BD Biosciences, San Diego, CA, USA	20 μL/10^6^ cells
HLA-DR (PE)	G46-6 (mouse IgG2a)	BD Biosciences, San Diego, CA, USA	20 μL/10^6^ cells
IL-4 (PE)	8D4-8 (mouse IgG1)	BD Pharmingen, San Jose, CA, USA	0.25 μg/10^6^ cells
IFN-γ (PE)	B27 (mouse IgG1)	BD Pharmingen, San Jose, CA, USA	0.5 μg/10^6^ cells
IL15 (PE)	34559 (mouse IgG1)	R&D Systems, Minneapolis, MN, USA	10 μL/10^6^ cells
CCL3 (PE)	93342 (mouse IgG2b)	R&D Systems, Minneapolis, MN, USA	20 μL/10^6^ cells
CCL17 (PE)	54015 (mouse IgG1)	R&D Systems, Minneapolis, MN, USA	20 μL/10^6^ cells
CCL22 (PE)	57203 (mouse IgG2b)	R&D Systems, Minneapolis, MN, USA	20 μL/10^6^ cells
**Secondary Antibodies**			
Anti-mouse IgG Alexa Fluor 488	goat IgG (H+L)	Molecular Probes, OR, USA	1:500
Anti-rabbit IgG Alexa Fluor 594	goat IgG(H+L)	Molecular Probes, OR, USA	1:500
**Control Antibodies**			
IgG1 (FITC, PE)	MOPC-21 (mouse IgG1)	BD Biosciences, San Diego, CA, USA	20 μL/10^6^ cells
IgG2a (FITC, PE)	G155-178 (mouse IgG2a)	BD Biosciences, San Diego, CA, USA	20 μL/10^6^ cells
IgG2b (FITC, PE)	27–35 (mouse IgG2b)	BD Biosciences, San Diego, CA, USA	20 μL/10^6^ cells
IgG2a (PE)	R35-95 (rat IgG2a)	BD Biosciences, San Diego, CA, USA	20 μL/10^6^ cells
IgG1 unconjugated	X40 (mouse IgG1)	BD Pharmingen, San Diego, CA, USA	dilution 1:100

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
