# Peer review of "Human Decidual CD1a+ Dendritic Cells Undergo Functional Maturation Program Mediated by Gp96"

_ijms, 2023, doi:10.3390/ijms24032278_

Round 1

Reviewer 1 Report

Reviewer Comments to Author

The article entitled "Human decidual CD1a+ dendritic cells undergo functional maturation program mediated by Gp96" indicates that gp96 specifically binds CD91 and TLR4 and cause proinflammatory maturation program in normal early pregnancy decidual CD1a+ DCs mediated by increased production of INF-γ and IL-15 in the context of Th1 cytokine/chemokine domination.Overall, the article is well written, but there are several issues as described below:

1. What does the (ref) in the article mean? Please mark the specific number of references.

2. In the result part: “Quantification of the immunohistological labels showed that the number of cytokeratin+ cells per mm2 was significantly lower in M.A compared to NP and B.O .....”Please note that the MA andBO format is available.

3. “3.4. Gp96 incude maturation of decidual CD1a+ DCs”. Author mentioned only the frequency of maturation CD83 marker increased in CD1a+ cells at the concentration of 1000 ng/ml of gp96, so the result may be controversial.

4. In the discussion part: “Gp96 in dose dependent manner also increased remarkably the percentages of IL-15 and IFN expressing CD1a+ DCs bias proinflammatory microenviroment domination.” Author could complement the role of IFN.

Author Response

The article entitled "Human decidual CD1a+ dendritic cells undergo functional maturation program mediated by Gp96" indicates that gp96 specifically binds CD91 and TLR4 and cause proinflammatory maturation program in normal early pregnancy decidual CD1a+ DCs mediated by increased production of INF-γ and IL-15 in the context of Th1 cytokine/chemokine domination. Overall, the article is well written, but there are several issues as described below:

We thank the Reviewer for his/her appreciation of our efforts. We have tackled the critical issues as detailed in the following point-by-point reply.

  1. What does the “(ref)” in the article mean? Please mark the specific number of references.

According to your suggestions, we apologize for missing references and we correct in the new version of manuscript and the specific numbers of references are marked in the red.

  1. In the result part: “Quantification of the immunohistological labels showed that the number of cytokeratin+ cells per mm2 was significantly lower in M.A compared to NP and B.O .....”Please note that the MA and BO format is available.

Following your suggestion, we correct all sentences with MA instead of M.A and the same was done with for blighted ovum BO. In the abstract and in the figure we have already mentioned that we will use the abbreviation for missed abortion MA and for blighted ovum BO.

  1. “3.4. Gp96 incude maturation of decidual CD1a+ DCs”. Author mentioned only the frequency of maturation CD83 marker increased in CD1a+ cells at the concentration of 1000 ng/ml of gp96, so the result may be controversial.

In line with literature, we found increased expression levels of CD83 after stimulation with gp96 on decidual CD1a+ DCs in vitro. Some research groups were stimulated mouse and human DC by addition of 30- 100 micro/ml gp96 and some others used low doses of gp96 that we used in our experimental model. So maybe for CD83 marker to obtained dose dependent manner upregulation should be used higher doses of gp96. Our model was set with lower doses as used in different research group. Also we teste our gp96 with the limulus amoebocyte lysate (LAL) test so we were sure that is not contaminated with LPS during experimental manipulation.

  1. In the discussion part: “Gp96 in dose dependent manner also increased remarkably the percentages of IL-15 and IFN expressing CD1a+ DCs bias proinflammatory microenviroment domination.” Author could complement the role of IFN.

Prompted by this comment, we have added in discussion part of manuscript marked in red the role of INF-gamma “Many of these IFNs are employed in normal pregnancy and development, as well as in defense against pathogens (40,41,42).  Aberrant expression of IFN-γ can alternate or induce aberrant activation of decidual NK cells during pregnancy that can lead to pregnancy complications. Our result indicates that increased concentration of gp96 at maternal-fetal interface maybe one of mechanism that could be a trigger for increased production of INF-γ.”

Following the suggestions of the reviewers we have critically read the manuscript and we ourselves have noted several shortcomings, which we corrected. We also restructure the “Discussion“ section in order to shape better in the new version of the manuscript according to reviewers suggestions. All of these changes are emphasized in the red font in the new version of the manuscript that can be easily seen.

Finally, we thank indeed the reviewers for the valuable suggestions, whose application has significantly improved the quality of the manuscript.

Reviewer 2 Report

The authors present an interesting work. A number of questions arose while reading:

1. Judging by the results of flow cytometry, there are quite a few dendritic cells in the decidua. I would like to clarify the technical details of obtaining a sufficient number of dendritic cells for analysis.

2. Did the authors test the obtained cultures for macrophage markers?

3. It should be noted that in the case of a rank analysis of variance, which the authors use for statistical analysis, it is necessary to use the Dunn test as a post-hoc test, and not the Mann-Whitney test.

Author Response

The authors present an interesting work. A number of questions arose while reading:

We thank the Reviewer for his/her appreciation of our efforts. We have tackled the critical issues as detailed in the following point-by-point reply.

  1. Judging by the results of flow cytometry, there are quite a few dendritic cells in the decidua. I would like to clarify the technical details of obtaining a sufficient number of dendritic cells for analysis.

We have investigated that issue and in our previously published data and manuscripts published from the other investigation groups, we determinate the optimal method to define number, distribution and functional properties of tiny population of decidual  CD1a+ cells among decidual mononuclear cells.  The phenotype of CD1a+ cells from freshly isolated DMC was analyzed by flow cytometry within region 1 (R1 gate) set for the CD1a+ events (back gating using CellQuestPro Software; Becton Dickinson) (Fig. 1a). This gate comprised the cells larger in size and of more complex granularity in comparison with the classical lymphocytes gate. In the representative sample, 2.59% of CD1a+ cells (R2 gate) were detected using antibodies of OKT-6 clone, in comparison with 1.38% of isotype-matched control staining (Fig. 1a). It goes along with 3.62±1.82% (mean ± S.D.) of CD1a+ cells found in freshly isolated DMC suspensions in 12 experiments performed (not shown).

Laskarin G, Redzović A, Rubesa Z, Mantovani A, Allavena P, Haller H, Vlastelić I, Rukavina D. Decidual natural killer cell tuning by autologous dendritic cells. Am J Reprod Immunol. 2008 May;59(5):433-45.

  1. Did the authors test the obtained cultures for macrophage markers?

We have investigated that issue in our previously published data where we as other groups found out that mostly decidual macrophages are skewed M2 macrophages phenotype in the uterine stroma to prevent rejection of the fetus and ultimately play roles in parturition. For example, compared with peripheral blood macrophages, the upregulated surface markers CD206, macrophage mannose receptor, CD209, and CD163 were found in systemic level of pregnant women as well as in the human pregnant uterus. 55-59 Moreover, decidual macrophages produce IL-10 and are the major source of IL-10 in the decidua, confirming their role in immunosuppression.

Laskarin G, Cupurdija K, Tokmadzic VS, Dorcic D, Dupor J, Juretic K, Strbo N, Crncic TB, Marchezi F, Allavena P, Mantovani A, Randic L, Rukavina D: The presence of functional mannose receptor on macrophages at the maternal-fetal interface. Hum Reprod 2005; 20: 1057– 1066.

Kammerer U, Eggert AO, Kapp M, McLellan AD, Geijtenbeek TB, Dietl J, van Kooyk Y, Kampgen E: Unique appearance of proliferating antigen-presenting cells expressing DC-SIGN (CD209) in the decidua of early human pregnancy. Am J Pathol 2003; 162: 887– 896.

Svensson J, Jenmalm MC, Matussek A, Geffers R, Berg G, Ernerudh J: Macrophages at the fetal-maternal interface express markers of alternative activation and are induced by M-CSF and IL-10. J Immunol 2011; 187: 3671– 3682.

Laskarin G, Redzović A, Rubesa Z, Mantovani A, Allavena P, Haller H, Vlastelić I, Rukavina D. Decidual natural killer cell tuning by autologous dendritic cells. Am J Reprod Immunol. 2008 May;59(5):433-45.

Laskarin G, Redzovic A, Vukelic P, Veljkovic D, Gulic T, Haller H, Rukavina D. Phenotype of NK cells and cytotoxic/apoptotic mediators expression in ectopic pregnancy. Am J Reprod Immunol. 2010 Nov;64(5):347-58. doi: 10.1111/j.1600-0897.2010.00844.x

  1. It should be noted that in the case of a rank analysis of variance, which the authors use for statistical analysis, it is necessary to use the Dunn test as a post-hoc test, and not the Mann-Whitney test.

Thank you for your precious suggestion that we missed. We apologize for this huge mistake; we did the Dunn test as post hoc test for these analyses. In the new version of manuscript we corrected it and tis marked in red.

Following the suggestions of the reviewers we have critically read the manuscript and we ourselves have noted several shortcomings, which we corrected. We apologize for spelling mistakes.

Finally, we thank indeed the reviewers for the valuable suggestions, whose application has significantly improved the quality of the manuscript.

Round 2

Reviewer 2 Report

Satisfactory answers have been received to all questions.